# DUAL DIFFUSION MODEL FOR ONE-SHOT HIGH-FIDELITY TALKING HEAD GENERATION

## ABSTRACT

One-shot audio-driven talking head generation is a significant task with applications in the movie industry and virtual avatars. However, existing methods have limitations in accurately capturing dynamic nuances within the mapping of audio-to-lip motion. Furthermore, GAN-based models for converting lip motion into pixel-level video often exhibit unstable training. To overcome these limitations, recent approaches based on diffusion models are proposed but still face issues such as time consumption and maintaining temporal consistency due to stochasticity. To circumvent these challenges, we introduce the following two modules: 1) AToM-Net, tasked with the generation of audio-to-motion pairs, and 2) MC-VDM, designed to produce high-quality image sequences corresponding to generated motion sequences reflecting a single identity image. Both modules are grounded in the framework of diffusion models. AToM-Net, with its inherent stochasticity akin to diffusion models, excels in capturing the subtleties of lip motion dynamics, avoiding the problem of mode collapse. MC-VDM solves the problems of the existing diffusion-based talking head by utilizing the efficient tri-plane based module. Our experiments conducted on the standard benchmark indicate that our model achieves performance that surpasses that of existing models.

## 1 INTRODUCTION

Audio-driven talking head generation is a challenging task that synthesizes a video with realistic lip movement corresponding to a given audio input. This cutting-edge technology has been widely studied due to its wide range of applications in diverse practical settings such as film production, digital avatars, etc.

Until recently, the majority of methodologies are based on Generative Adversarial Networks. Among them, previous works (KR et al., 2019; Prajwal et al., 2020) primarily focused on generating lip movements that synchronize well with the audio. However, given the inherently high-dimensional nature of image features in contrast to audio, it becomes challenging to precisely control lip movements based on audio information, so that they fail to generate natural and realistic videos. To mitigate this limitation, some methods propose the networks to utilize intermediate facial representation such as dense motion fields (Zhang et al., 2023) or 3D facial landmarks (Wang et al., 2021a;b). Since 3D facial models can explicitly represent the facial motion well, the network using 3D facial landmarks can generate more realistic videos with natural pose motion. Despite the implementation of innovative methods, these approaches exhibit significant limitations, which are the instability of the GAN learning process and the mode collapse in generated results. There are a few recent works (Shen et al., 2023; Stypułkowski et al., 2023) using the diffusion model to improve image quality and model generalization for talking head generation tasks. However, due to the extensive parameters of the diffusion model, the training process is computationally demanding, and sampling also takes a considerable amount of time. Moreover, since diffusion is characterized by its high diversity, they have difficulty in preserving temporal consistency compared to GAN-based model.

In this work, we divide our task into two-stage models to address these challenges. First, we propose the diffusion-based AToM-Net to generate the facial landmark corresponding to audio as the intermediate representation. We utilize explicit 3DMM coefficients to extract frontal facial landmarks, aiming to train the model to generate landmarks with accurate lip shapes by eliminating pose infor-

mation. To enhance lip synchronization performance, we present a transformer-based network that incorporates separate attention mechanisms, distinguishing between lip-related and non-lip-related regions. To tackle computational challenges caused by huge diffusion model, we employ a tri-plane approach to generate temporally coherent videos that align with both the reference image conditions and the motion predictions generated by AToM-Net. Under these designs, we introduce a diffusion-based model capable of generating high quality talking face videos from just a single-shot image for the first time. We evaluate our method on HDTF dataset. Experiments demonstrate that our proposed method surpasses the performance of previous talking head generation models, encompassing both GAN-based and Diffusion-based approaches. Furthermore, we offer comprehensive ablation studies for a thorough evaluation.

## 2 RELATED WORK

**Talking Head Generation**     Talking head video generation with lip movements aligned with audio inputs is a long-standing problem in computer vision. While earlier studies (Kumar et al., 2017; Suwajanakorn et al., 2017) focused on designing person-specific talking head networks for a single target individual, more recent studies (Prajwal et al., 2020) have explored unified reconstruction-based frameworks that can synthesize lip movements for any individual.

Wav2Lip (Prajwal et al., 2020) adopts an inpainting method to match the mouth movements with the audio inputs, leveraging the pretrained syncnet (Chung & Zisserman, 2017) as a lip sync expert to achieve high-quality results. However, it suffers from blurring artifacts, in which the reconstructed lip area and the original upper face area are not blended well. Furthermore, several studies (Zhou et al., 2021; Ji et al., 2022) have proposed methods to synthesize not only lip movements but also other visual components such as pose and emotions. MakeItTalk (Zhou et al., 2020) employs 3D landmarks to generate personalized head motions and expressions, whereas PC-AVS (Zhou et al., 2021) devises an implicit low-dimensional pose code to control lip movements and head poses respectively. GC-AVT (Liang et al., 2022) uses individual encoders for head pose, audio content, and emotional expression to decouple all of these factors. To capture the detail of facial motions, recent models (Zhang et al., 2021; Ren et al., 2021) utilize 3D Morphable Model parameters which consist of expression, pose, and identity. Sadtalker (Zhang et al., 2023) design the sub-networks that can independently generate lip expressions and head pose parameters with enhanced accuracy. While it successfully produces high-quality video, it is impossible to avoid the inherent limitations of GAN models.

Recently, DiffTalk (Shen et al., 2023) proposed the multi-shot talking face generation network utilizing the diffusion model for the first time. While it can generate high-quality talking face videos, it demands numerous computational resources to train and a huge inference time to generate videos. To address these challenges, we introduce a novel approach – the diffusion-based one shot talking face generation model. Our model operates efficiently, requiring minimal computational resources.

**Video Diffusion**     Diffusion models have demonstrated exceptional performance in generative tasks, causing a great impact in the field of computer vision. Especially, LDM (Rombach et al., 2022) proposed a model that performs the diffusion process in a lower-dimensional latent space, thus resolving the issue of high computational cost. Motivated by the impressive generative performance achieved by LDM, some models (Blattmann et al., 2023; Wu et al., 2022; Liu et al., 2023), which extend the pre-trained T2I diffusion model to the video task, have been developed. Although they exhibit impressive results with minimal computational resources, there are still certain limitations that need to be addressed. Because the results are generated frame by frame, they have difficulties in achieving temporal consistency and generating smooth videos at a high frame rate. Additionally, these models struggle with producing realistic images, such as human faces, as they are influenced by the bias inherent in the pre-trained diffusion model, which typically generates synthetic images.

On the other hand, several methodologies (Ho et al., 2022; Yu et al., 2023; Hu et al., 2023; Wang et al., 2023) have been proposed for training diffusion models from scratch using video datasets. These models have shown the performance to produce greater outcomes that exhibit temporal consistency and authenticity. However, they still face limitations related to memory inefficiencies and controllability. To overcome the computational problem, PVDM (Yu et al., 2023) utilizes the diffu-

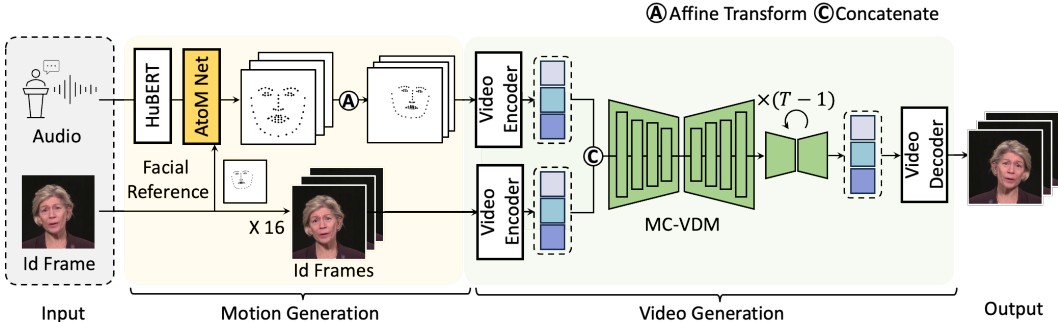

Figure 1: **Overview of the proposed talking face video generation model.** We synthesize a audio-driven talking portrait in two stages. First, motion generation aims to generate lip-syncronized frontal face. Second, Video generation create natural results using synthesized frontal landmarks and identity conditions.

sion model for video generation, focusing on generative modeling in low-dimensional latent space, but it has a critical limitation that only produces unconditional results. LaMD (Hu et al., 2023) solves Image-to-Video (I2V) and Text-Image-to-Video (TI2V) tasks by using the autoencoder module which incorporates a 3D CNN encoder and a multi-scale 2D CNN encoder. LEO (Wang et al., 2023) proposes a diffusion model for human video generation by representing motion as a sequence of flow maps. By doing so, it can produce both image-conditioned and unconditioned videos. Our novel diffusion model based on the tri-plane structure can efficiently generate smoothed talking face video corresponding to a one-shot identity image and the sequences of landmarks produced by our audio-to-land model.

## 3 METHODOLOGY

### 3.1 OVERVIEW.

Our entire model consists of two parts as shown in Fig.1. As illustrated in Fig.2, Audio-To-Motion Generator, named AToM-Net is composed of a diffusion model, which reconstructs 3D landmarks by incorporating audio features and an initial 3D landmark as conditions. Especially, we propose the transformer based generator that separates lip and non-lip regions to improve lip-sync performance. Then, we train the diffusion-based Motion-to-Video network, named MC-VDM to generate realistic talking face video by utilizing predicted landmarks and a single identity image in Fig.3. In this section, we explain the proposed AToM-Net approach and MC-VDM approach in detail.

### 3.2 AUDIO-TO-MOTION GENERATION

**Audio and Motion Representation** Generating the landmark of talking portraits from audio is challenging task, given the ill-posed property of diverse head pose. Many early approaches follow the pre-estimated structural information, using a face landmark detector with a head pose estimator (Zhou et al., 2020; Zhang et al., 2021; Ren et al., 2021). However, they tend to synthesize unrealistic lip-sync motion due to the high dimensionality of facial movement. To achieve faithful talking face generation with lip-syncing, we introduce a diffusion-based Audio-to-Motion Generator to perform expressive lip motion sequences.

Our method utilize a diffusion models (DMs) with strong audio feature extractor and 3D facial morphable model (3DMM). To obtain the meaningful information from the raw waveform of the speech signal, we extract audio features using HuBERT which have demonstrated a state-of-the-art performance on automatic speech recognition (ASR). Regarding the motion representation, we take 68 face landmark from the reconstructed 3D face mesh and use them as facial motion representations.

With 3DMM, a 3D face mesh $M$ can be represented as a linear combination of identity and expression basis.

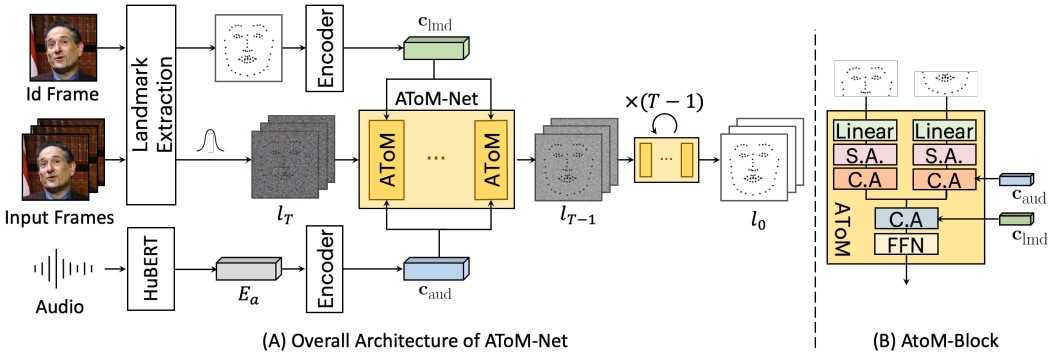

Figure 2: **Audio-to-Landmark(AToM) pipeline overview** (Left) Our AToM learn to denoise facial landmark sequence from diffusion process. The model aims to generate the corresponding lip sync landmark according to the audio feature and initial lip condition. (Right) Our AToM module splits the upper and lower parts of the face so that the audio only responds to relevant keypoints.

$$M = \overline{M} + M_{\text{id}}\alpha_{\text{id}} + M_{\text{exp}}\alpha_{\text{exp}}, \tag{1}$$

where $\overline{M}$ is the mean face shape; $M_{\text{id}}$ and $M_{\text{exp}}$ denotes the identity and expression matrices; $\alpha_{\text{id}}$ and $\alpha_{\text{exp}}$ denotes the identity and expression parameters. To separate the movement of the lip motion from the pose, we transformed the posed face keypoints into a frontal face keypoints as follows:

$$LM_{3D} = \{(M - \overline{M})_i | i \in I\}, \tag{2}$$

Here, $LM_{3D} \in \mathbb{R}^{68 \times 3}$ represent 3D face landmarks, $M$ and $\overline{M}$ correspond to the 3DMM mesh and mean face mesh.

**Audio-to-Motion Generator**   Given a single reference image **r**, our goal is generating synchronized landmarks according to the audio. To facilitate robust lip synchronization, we leverage a transformer-based architecture where audio features $\mathbf{c}_{\text{aud}}$ and reference landmarks $\mathbf{c}_{\text{lmd}}$ are incorporated into timestep $t$. Specifically, we design **AToM-Net** to disentangle a face keypoints into two components of lip-related and non-related parts. By categorizing them into two parts, we can improve lip synchronization quality with audio-related condition. Additionally, we utilize a feature-wise linear modulation block (FiLM) to effectively handle the conditioning information. Overall architecture of **AToM-Net** is shown in Fig.2.

**Training Process**   Following the DDPM (Ho et al., 2020) definition, we perform diffusion process on the facial motion sequences. Given a facial motion sequence $\boldsymbol{l_0} \sim p_{data}(\boldsymbol{l_0})$, we train diffusion models to generate landmark sequence from Gaussian distribution $\boldsymbol{l}_T \sim \mathcal{N}(0, \boldsymbol{I})$. The forward noising process is gradually adding noise to $\boldsymbol{l_0}$ according to a predefined variance schedule:

$$q(\boldsymbol{l}_t | \boldsymbol{l_0}) \sim \mathcal{N}(\sqrt{\bar{\alpha}_t}\boldsymbol{l_0}, (1 - \bar{\alpha}_t)\boldsymbol{I}), \tag{3}$$

where $\bar{\alpha}_t$ are monotonically decreasing variance schedule and the data point $x_T$ becomes Gaussian noise as $\bar{\alpha}_t$ get closer to 0. In the reverse process, diffusion model learn a backward process by estimating $\hat{\boldsymbol{l}}_\theta(\boldsymbol{x}_t, t, \mathbf{a}, \boldsymbol{l}) \approx \boldsymbol{l}$. Where we denote $\theta$ as model parameters and $t$ as timestep. In the end, we optimize $\theta$ with a simplified version of objective as follow:

$$\mathcal{L}_{\text{simple}} = \mathbb{E}_{\boldsymbol{l},t}\left[\|\boldsymbol{l} - \hat{\boldsymbol{l}}_\theta(\boldsymbol{l}_t, t, \mathbf{c}_{\text{aud}}, \mathbf{c}_{\text{lmd}})\|_2^2\right]. \tag{4}$$

### 3.3 MOTION-CONDITIONAL VIDEO DIFFUSION MODEL

After generating landmark sequences synchronized with the audio, we synthesize a pixel-level frame sequence using given landmark sequence as a spatial condition. Previous methods (Stypułkowski et al., 2023; Shen et al., 2023) whose backbone image generation models are diffusion models (Ho

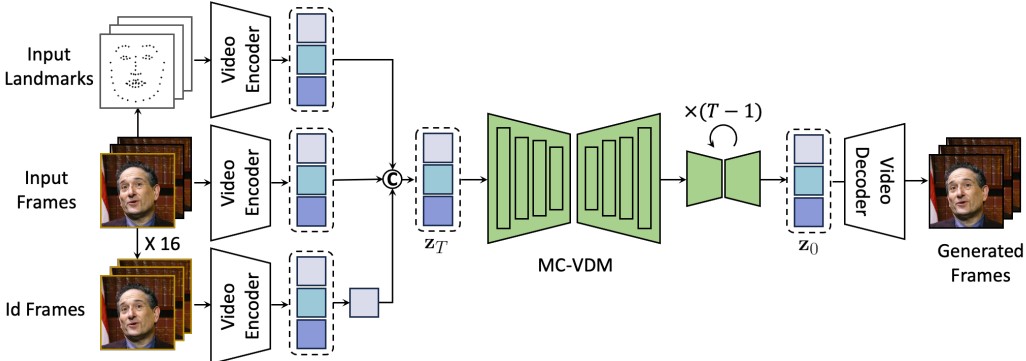

Figure 3: **Motion-Conditional Video Diffusion model(MC-VDM) pipeline overview.** MC-VDM takes the landmark and reference condition as input and synthesizes the corresponding video frame.

et al., 2020; Rombach et al., 2022) have struggled to maintain consistency in portraying a single person across a video due to stochasticity caused by denoising processes. (Shen et al., 2023) employed frame-by-frame generation where an upper face ground truth is provided for each frame. To address this challenge, we propose the Motion-Conditional Video Diffusion Model, dubbed **MC-VDM** which generate temporal consistent frames corresponding given facial condition sequences.

Our module **MC-VDM**, based on PVDM (Yu et al., 2023) that represents video voxels using three 2D latents, generates multiple frames simultaneously. This approach allows for the preservation of identity while mitigating common issues associated with diffusion-based models like jittering. Exploiting the temporal and spatial redundancy of video is highly effective for talking head generation due to the nature of the task, which deals with datasets that have a static background and consistent content.

**Facial motion alignment** Since AToM-Net is trained with a dataset pre-processed to align with the mean face of the 3DMM, the generated motion sequence is a facial motion fixed at the position and pose of the 3DM template mean face.

The 3D facial landmarks generated by the audio-to-motion module are modified to match the face position, size, and pose of the driving video. Specifically, 2D landmarks are extracted from each frame in the pose driving video and used with generated 3D facial landmarks to calculate rotation and translation matrix using the POS algorithm. An affine transformation is then applied to align a sequence of motion landmarks, originally associated with the mean face of a 3DMM, with the face in the driving video, adjusting for its position, pose, and size.

**Conditional video diffusion model for talking head** MC-VDM is comprised of two main components: a video autoencoder and a denoising Unet model. The video autoencoder consists of an encoder $\mathcal{E}_\phi$ and a decoder $\mathcal{D}_\psi$ (Bertasius et al., 2021) with projection networks (Vaswani et al., 2017).

We utilize samples of clips $\mathbf{x} \in \mathbb{R}^{S \times H \times W}$ composed with S frames from data distribution $p_{\text{data}}(\mathbf{x})$. The video encoder $\mathcal{E}_\phi$ compress each clips into three 2D latents $\mathbf{zx} = [\mathbf{zx^{xy}}, \mathbf{zx^{xt}}, \mathbf{zx^{yt}}]$. Denoising module learns the distribution of dataset while learning the process of denoising noisy projected latents. During the training process, the video diffusion model learns the distribution within the dataset. Therefore, being able to generate unconditional videos that accurately depict natural lip and head movement between neighboring frames. Moreover, this can be achieved efficiently without compromising on quality by employing three 2D-latent representations.

However, we cannot control over the content appeared in synthesized clips, which is the most important factor within the talking face generation. To overcome this limitation, we introduce practical conditioning approach, dubbed MC-VDM. We inject the identity visual ques $\mathbf{r} \in \mathbb{R}^{1 \times H \times W}$ and motion sequences conditions $\boldsymbol{l} \in \mathbb{R}^{S \times H \times W}$ into the video diffusion model.To obtain the conditioning latents, we use encoder $\mathcal{E}_\phi$ to compress a reference frame $\mathbf{r} \in \mathbb{R}^{1 \times H \times W}$ into reference latents $\mathbf{zr}$ = $[\mathbf{zr^{xy}}, \mathbf{zr^{xt}}, \mathbf{zr^{yt}}]$. Then, we use the latent $\mathbf{zr^{xy}}$ extracted through our video encoder as the identity condition. Additionally, encoder $\mathcal{E}_\phi$ encode facial landmark frames $\boldsymbol{l} \in \mathbb{R}^{S \times H \times w}$ into $\mathbf{zl} =$

$[\mathbf{z}l^{\mathbf{xy}}, \mathbf{z}l^{\mathbf{xt}}, \mathbf{z}l^{\mathbf{yt}}]$. We concat extracted $\mathbf{zx}, \mathbf{z}l, \mathbf{zr}^{\mathbf{xy}}$ from a pre-trained video encoder latents making $\mathbf{z} = [\mathbf{zx}, \mathbf{z}l, \mathbf{zr}^{\mathbf{xy}}]$. By leveraging the latent $\mathbf{zr}^{\mathbf{s}}$ which capture the identity information, we can effectively guide the generation of subsequent frames and ensure coherence throughout the synthesized video sequence.

**Training Process** MC-VDM model involving two autoencoder modules is trained under a two-stage training process, respectively. The first auto encoder compress the cubic representation of video into image-like three 2D latents. $\mathcal{E}_{\phi} : \mathcal{X} \rightarrow \mathcal{Z}$ with $\mathcal{E}_{\phi}(\mathbf{x}) = \mathbf{z}$ and a decoder $\mathcal{D}_{\psi} : \mathcal{Z} \rightarrow \mathcal{X}$ with $\mathcal{D}_{\phi}(\mathbf{z}) = \tilde{\mathbf{x}}$ so that $\tilde{\mathbf{x}}$ becomes $\mathbf{x}$. For training, the sum of objective functions: pixel-level reconstruction loss and the negative of perceptual similarity are used. The corresponding objective can be formulated as:

$$\mathcal{L}_{\text{rec}} = \mathbb{E}_{\mathbf{x} \sim \mathcal{X}} \left[ \|\mathbf{x} - \tilde{\mathbf{x}}\|_1 \right]. \tag{5}$$

$$\mathcal{L}_{\text{LPIPS}} = \mathbb{E}_{\mathbf{x} \sim \mathcal{X}} \left[ \|\phi(\mathbf{x}) - \phi(\tilde{\mathbf{x}})\|_1 \right]. \tag{6}$$

where $\phi$ denotes the perceptual feature extractor.

$$\mathcal{L} = \lambda_1 \mathcal{L}_{\text{rec}} + \lambda_2 \mathcal{L}_{\text{LPIPS}}. \tag{7}$$

where we choose $\lambda_1 = \lambda_2 = 1$ in our experiments.

The second denoising autoencoder learns the data distribution within $p_{\text{data}}(\mathbf{z})$ by gradual denoising process, making Gaussian prior distribution to $p_{\text{data}}(\mathbf{z})$. For training, Noise prediction objective is used. More specifically, the training process of our motion controllable image-to-video diffusion model is formalized as the following:

$$\mathbb{E}_{\boldsymbol{\epsilon}, t} \left[ \lambda \|\boldsymbol{\epsilon} - \boldsymbol{\epsilon}_{\boldsymbol{\theta}}(\mathbf{z}_t, t)\|_2 \right].$$

where $\mathbf{z}_0 = [\mathbf{zx}, \mathbf{z}l, \mathbf{zr}^{\mathbf{xy}}] = \mathcal{E}(\mathbf{x}), \mathbf{z}_t = \sqrt{\bar{\alpha}_t} \mathbf{z}_0 + \sqrt{1 - \bar{\alpha}_t} \boldsymbol{\epsilon}$.

## 4 EXPERIMENTS

### 4.1 EXPERIMENT DETAILS.

**Datasets** We use LRES3-TED (Afouras et al., 2018) to train our motion generator, AToM-Net and HDTF (Zhang et al., 2021) dataset to train video generator, MC-VDM, respectively. LRS3-TED contains 400 hours of TED videos with a large lip reading corpus. We train AToM-Net by extracting video frames at 25fps and audio at 16000 sampling rate. As for the MC-VDM, we choose random 312 videos for training and the remaining 98 videos for testing from the HDTF dataset.

**Implementation details** We train our AToM-Net and MC-VDM separately on 1 NVIDIA RTX 3090 GPU. For AtoM-Net, it takes 300k iterations steps for training. For MC-VDM, we train the model for 600k iterations which takes about 96 hours. The hyperparameters needed for each model can be found in the Appendix.

**Compared Models** We perform method comparisons with several state-of-the-art methods: 1) 2D-based talking head generation (Audio2Head (Wang et al., 2021a), PC-AVS (Zhou et al., 2021), Wav2Lip (Prajwal et al., 2020)) ; 2) MakeItTalk (Zhou et al., 2020), which utilize interemediate representation ; 3) SadTalker (Zhang et al., 2023), which is flow-based approach ; 4) DiffTalk (Shen et al., 2023), which is diffusion-based method. Moreover, to validate the performance of AtoM-Net, we employ the variational motion generator from GeneFace (Ye et al., 2023) as a comparative model.

**Evaluation Metrics** We assess the performance of our proposed method using several metrics that have been frequently utilized in previous research. Specifically, we employ the **FID** ($\downarrow$) (Heusel et al., 2017; Seitzer, 2020) metric to gauge the image quality and **CPBD** ($\uparrow$) (Narvekar & Karam, 2011) to evaluate the sharpness of the generated frames. Moreover, to examine the identity preservation, we conducted the **CSIM**($\uparrow$) (cosine similarity). For assessing the lip synchronization quality, which is essential for the audio-driven talking head generation, we conducted SyncNet (Chung & Zisserman, 2017) score. **LSE-D**($\downarrow$) and **LSE-C**($\uparrow$), higher is better) evaluate the audio-lip synchronization quality and estimate the accuracy of mouth shape by utilizing Syncnet (Chung &

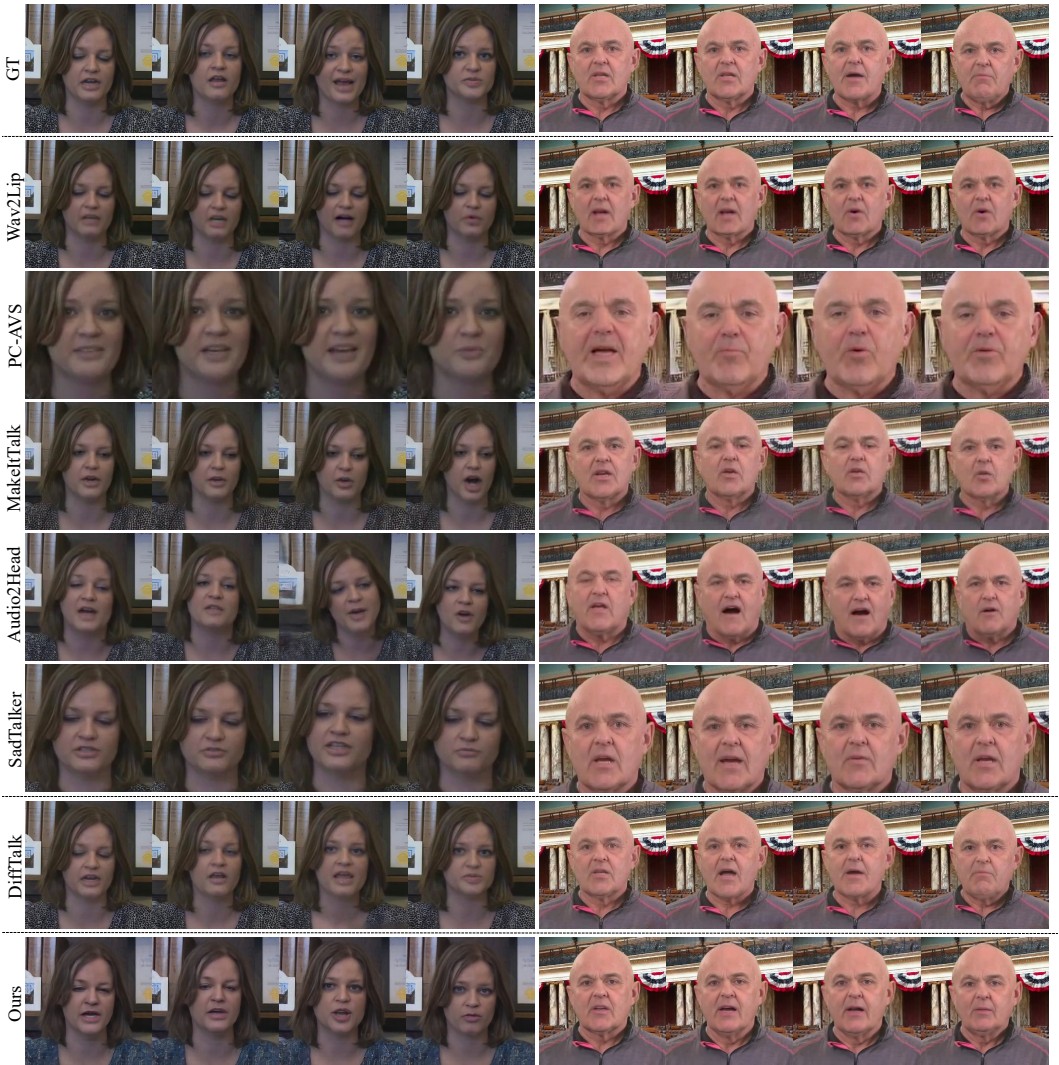

Figure 4: **Qualitative comparison on different reference images and pose.**

Zisserman, 2017). LSE-D quantifies synchronization by calculating the L2 distance between audio and video features and LSE-C focuses on the confidence score associated with the time offset of audio-video synchronization. Finally, **LMD**($\downarrow$) is L1 loss that measures the accuracy of generate lip movements by computing distance between each corresponding pairs of landmarks on real video and synthesized video.

### 4.2 QUANTITATIVE RESULTS

We conduct quantitative evaluation of our model with other talking head generation methods on HDTF (Zhang et al., 2021). As presented in Tab.1, our model outperforms the current state-of-the-art methods in terms of various metrics which measure the video quality. These results demonstrate that our MC-VDM is faithfully designed to generate natural and high-fidelity images while preserving identity effectively. Moreover, even in the absence of SyncNet, we achieve competitive scores when compared to previous methods that employ SyncNet in LSE-C and LSE-D, metrics quantifying the similarity between lip and audio features. Especially, Our superiority in prossessing efficiency is shown in Tab.2. Train time is measured with a single NVIDIA 3090Ti 24GB GPU. and inference time. Inference time is the time it takes to generate a 5-second video of 256 resolution. We achieve better performance similar to Difftalk while having significantly faster training and inference times.

Table 1: **Comparison with the state-of-the-art method on HDTF dataset.** Wav2Lip and DiffTalk excel in video quality because they only generate the lip region, leaving other areas of the ground-truth unchanged.

| Method | Lip Synchronization | | Video Quality | | |
|---|---|---|---|---|---|
| | LSE-C↑ | LSE-D↓ | FID↓ | CSIM↑ | CPBD↑ |
| Real Video | 8.211 | 6.982 | 0.000 | 1.000 | 0.428 |
| Wav2Lip  (Prajwal et al., 2020) | **9.471** | **5.857** | 33.298 | 0.747 | 0.349 |
| PC-AVS  (Zhou et al., 2021) | 8.680 | 6.613 | 117.848 | 0.478 | 0.294 |
| MakeItTalk (Zhou et al., 2020) | 4.387 | 10.229 | 74.262 | 0.713 | 0.433 |
| Audio2Head (Wang et al., 2021a) | 7.213 | 7.496 | 63.755 | 0.582 | 0.373 |
| SadTalker (Zhang et al., 2023) | 7.123 | 7.854 | 48.480 | 0.719 | **0.446** |
| DiffTalk (Shen et al., 2023) | 5.745 | 7.609 | 84.181 | 0.592 | 0.524 |
| Ours | 5.295 | 8.096 | **33.089** | **0.739** | 0.406 |

Furthermore, We compare our landmark generator AToM-Net with audio-driven landmark generation methods, Gene-Face (Ye et al., 2023). As shown in Tab.3, AToM-Net outperforms GeneFace in LMD. Also, we achieve competitive performance in Lip-sync metric even without using SyncNet. Lip-sync metrics even though we. GeneFace introduce a variational motion generator designed to produce precise and expressive facial landmark. They employ a Flow-based generative model to establish a complex, time-dependent distribution as the prior of the VAE. Unlike Flow-based method, our transformer-based

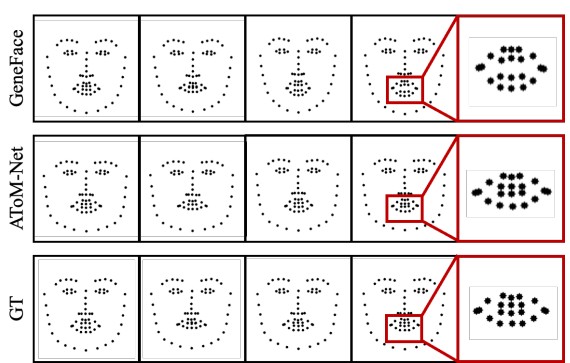

Figure 5: **Qualitative comparison with GeneFace on Landmark.**

diffusion model are learned to generate landmark sequence from Gaussian noise given the initial face landmark and input audio. Consequently, AToM-Net excels in efficiently capturing long-range relationships between audio and motion, which proves beneficial for the attention mechanism within the transformer architecture.

### 4.3 QUALITATIVE RESULTS

We illustrate visual results for Tab.1 in Fig.4. We demonstrate that our model excels in producing high-quality video results that closely resemble the target video, surpassing the performance of other models. Wav2Lip (Prajwal et al., 2020) and PC-AVS (Zhou et al., 2021) demonstrate precise lip synchronization but struggle with image quality, often resulting in blurriness. Conversely, Au-

Table 2: training and inference time with DiffTalk  (Shen et al., 2023).

| Method | Train | Inference |
|---|---|---|
| DiffTalk | 120h | 875s |
| Ours | **48h** | **125s** |

dio2Head (Wang et al., 2021a) and MakeItTalk (Zhou et al., 2020) show inaccuracies in lip synchronization and produce distorted outcomes. Sadtalker (Zhang et al., 2023) exhibits clumsy movements in various facial motions, including eye blink and pose, while Difftalk (Shen et al., 2023) exhibits jittering issues in the lower region, leading to reduced image quality. Thanks to the introduction of our innovative diffusion-based MC-VDM, which jointly considers the identity image and landmarks, our model achieves the capability to produce videos that maintain stability and identity coherence more effectively.

We also show landmark comparison with GeneFace (Ye et al., 2023) in Fig.5.Consistent with the quantitative findings, our AtoM-Net produces more precise lip landmarks. Notably, unlike Gene-

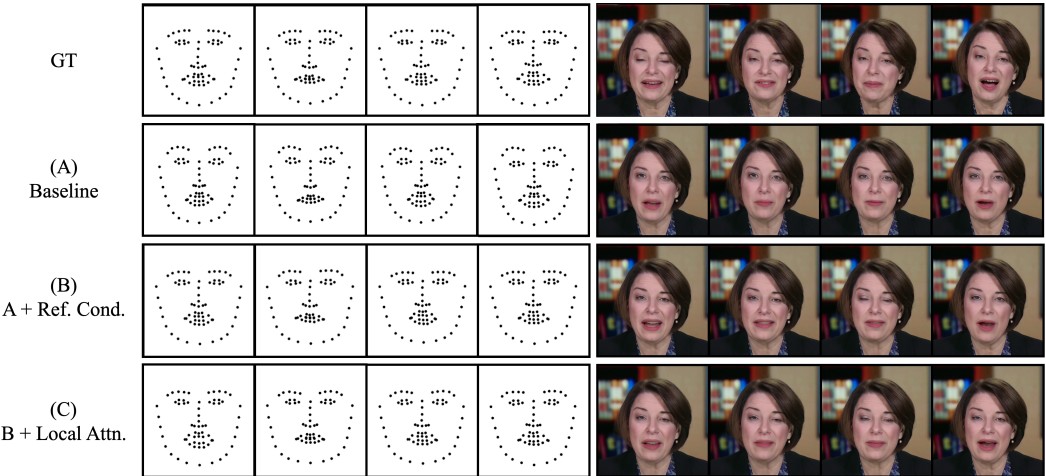

Figure 6: **Ablation study with visual results.** The mouth shapes are same among results but not synced with pose source.

face, our model is capable of generating personalized landmarks that correspond to a single reference image, eliminating the necessity for additional networks.

## 4.4 ABLATION STUDY

The advantage derived from our AtoM-Net is the accurate facial movement. As shown in Tab.3 and Fig.6, we observe that prediction accuracy improved as audio and facial keypoint features were disentangled. Especially, the baseline method that learns generating landmark sequence by simply concatenating face keypoints and audio fails to preserve identity and yields poor results. As for reference-based method, lip-synchronization performance was improved by injecting the starting point of the landmark which reflect the personality of the reference image. Lastly, our audio-to-landmark method achieve highest lip-synchronization score with disentanglement for audio-correlated and uncorrelated motion, ensuring the plausible landmark animation.

Table 3: Ablation Study for AToM-Net.

| | Method | LMD ↓ | Lip-sync ↑ |
|---|---|---|---|
| | Geneface | 137.52 | 0.339 |
| A | Baseline | 159.47 | 0.316 |
| B | + Ref. Cond. | 101.15 | 0.311 |
| C | + Local Attn. | **97.16** | **0.342** |

## 5 CONCLUSION

In this research, we present a two-stage diffusion model for generating talking heads from a one-shot input image. To create synchronized facial landmark sequences from audio, we introduce AToM-Net, a network comprising transformers specialized for lip-related and non-lip areas. Additionally, we propose MC-VDM, designed to generate temporally consistent image sequences that align seamlessly with both motion sequences and a single identity image, utilizing the inherent stochastic properties of the diffusion model. Our experiments clearly show that our model outperforms previous talking head synthesis models, including those based on GANs and diffusion-based techniques. Especially, as demonstrated in our qualitative assessment, owing to the stability and remarkable fidelity offered by diffusion models, our approach excels in producing synthetic videos of high quality. Furthermore, given our ability to generate precise personalized motions, our model offers a significant advantage in its applicability across a wide range of domains, with particular relevance to the animation industry.

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
