# OpenReview forum: "Dual Diffusion Model for One-Shot High-Fidelity Talking Head Generation"
_ICLR.cc/2024/Conference — ICLR 2024 Conference Withdrawn Submission_

### Official Review · Reviewer_FJie · 2023-10-25

**Soundness:** 2 fair
**Presentation:** 1 poor
**Contribution:** 2 fair
**Rating:** 3
**Confidence:** 4

**Summary:**

This paper presents a diffusion-based talking head generator.  The model first generates a sequence of landmarks from the audio, then these landmarks and an identity image are used with a conditional diffusion model to create the resulting image sequence.  To ensure consistency in the generated image sequence, blocks of image frames are generated together rather than frames generated individually.

**Strengths:**

+ Talking head generation is an important and challenging problem as viewers are highly sensitive to inconsistencies in lip motion.
+ Open datasets are used.
+ The approach seems sensible.  Controlling lip motion independently of non-speech facial motion allows the network to focus on what is important for speech, and helps avoid spurious correlations that might otherwise be captured.  Also, the use of 3D landmarks as an intermediate representation allows for effective pose normalization.

**Weaknesses:**

- The main weakness of the paper is missing important details about the architecture, the training, and any hyperparameters, which makes reproducing the work impossible.
- The paper states this is the first time one-shot diffusion models have been used for modeling talking faces.  However, see this work:  Diffused Heads: Diffusion Models Beat GANs on Talking-Face Generation.
- The abstract reads like the approach requires a single image of a person and some accompanying audio to generate talking head sequences.  However, the method also requires a sequence of landmarks to drive the pose of the talking head, as the speech motion generation is all aligned to the mean of the 3DMM.
- There is no accompanying video, which is the only way to gauge the quality of a talking head sequence.  The static frames in Figure 4 are insufficient — and this figure suggests problems with the generated lip shapes to me.  For the speaker in the lefthand images:  the third frame for your method has the lips too rounded, and the lip pursing in the fourth frame is under articulated.  For righthand speaker:  the inner mouth is wrong in the second frame, and the lip closure is missed in the fourth frame.
- There is no subjective / perceptual study to qualitatively gauge the quality of the generated lip motions.

**Questions:**

- If you wish to generate speech for a specific speaker in a one-shot fashion, how do you know how to generate the correct inner mouth if the reference frame has the mouth closed?  In addition, how does your model know how to capture the idiosyncrasies for a given speaker if you have not seen their specific speaking style before?  For example, in the extreme case that a speaker speaks out of one side of their mouth?
- Why are only 68 landmarks extracted from the broader 3DMM?  Why not use the full mesh?
- In equation (2), the difference between a mesh and the mean of the model does not transform “the posed face keypoints into a frontal face keypoints” — to do this you need to properly align (using rotations and scaling).  Also, is i in this sequence a frame index?
- In the description of the Audio-to-Motion Generation, you refer to a feature-wise linear modulation block (FiLM).  This does not appear to be labelled in the architecture diagrams, and there is no detail specifically about this.
- Sequences of frames are generated simultaneously to ensure that identity is consistent across frames — you still need to be concerned with identity across blocks of frames though.  This is not discussed.
- A driving video is required from which the pose is extracted for the generated sequence.  Is there any consideration for the consistency for the head pose that accompanies the speech?  Viewer can be sensitive to mismatches in head pose and speech (e.g., timing, rhythm, etc.).
- For the conditional video diffusion, three 2D latents are introduced — what are these representing?
- The latents zx represent S frames, but zr a single frame.  How is this mismatched handled?
- None of the loss terms related to speech information specifically.  This is surprising.  For example, Equation (5) is encouraging the images to be similar, but in a global sense.  Errors in the cheeks (or planar regions) are less significant the say lips almost being closed when they should actually be closed.
- Why was a user study not run to measure the quality of the speech generation?

---

### Official Review · Reviewer_KsG2 · 2023-10-30

**Soundness:** 2 fair
**Presentation:** 2 fair
**Contribution:** 2 fair
**Rating:** 5
**Confidence:** 4

**Summary:**

This paper presents a two-stage diffusion model for one-shot talking head generation. They use two diffusion models sequentially: (1) a HuBERT-conditioned DDPM for audio-to-motion prediction; (2) a landmark-conditioned PVDM for motion-to-video rendering. It contributions are twofolds: (1) it is the first work that utilize diffusion model for facial motion prediction; (2) it improves the temporal smoothness and system efficiency of the diffusion model for talking head generation. Compared with previous diffusion-based method, DiffTalk, it reduces the training time from 120h to 48h, and inference time from 875s to 125s. The paper writing is clear and is easy to follow.

However, the novelty is relatively limited, and the system efficiency still seems unpractical in real-world applications. Besides, there is demo video available in the initial submission, which is necessary to assess the quality of a video generation paper. In summary, I tend to give a rate of 5.

**Strengths:**

- It well utilizes two different diffusion models, DDPM and PVDM for talking face generation.
- It is the first work that utilizes diffusion model for facial motion prediction;
- By using PVDM, it improves the temporal smoothness and system efficiency of the diffusion model for talking head generation.

**Weaknesses:**

- The novelty is limited, both of DDPM used in audio-to-motion stage and PVDM used in motion-conditioned video generation stage are well-known models.
- No demo videos are available.

**Questions:**

- In Figure 3, it seems that several input frames are required for video rendering. This raises a concern that is it really a one-shot method?
- The author could provide demo videos for qualitative evaluation.

---

### Official Review · Reviewer_qQ9o · 2023-11-01

**Soundness:** 3 good
**Presentation:** 3 good
**Contribution:** 2 fair
**Rating:** 5
**Confidence:** 2

**Summary:**

This paper proposed a diffusion model based approach to synthesize full-face lip motion video conditioned on audio and a single facial image. The proposed method consists of two major components. The first one called AToM-Net is an audio-to-motion generation network that utilize HuBERT model and 3DMM to generate 3D facial landmarks given audio signals. The second one called MC-VDM takes the generated 3D landmarks and reference facial image to generate a sequence of frames with the same identity as the reference image and lip motion encoded by the landmarks. Experimental evaluation on benchmark dataset shows the proposed method has competitive temporal consistency and video quality compared to other state-of-the-art methods.

**Strengths:**

1. The paper proposed a novel approach to synthesize facial video with realistic lip motion conditioned on audio signal and a reference face image. The method has the advantage of diffusion model in high realism while keeping the temporal and identity consistency within a sequence of images.
2. The proposed synthesis framework has improved efficiency in both training and inference time compared to most recent diffusion model based approach.
3. Both quantitative and qualitative results shown the proposed method have competitive performance compared to state-of-the-art talking head generation methods.

**Weaknesses:**

1. Although the proposed method is intuitive and conceptually appealing in the sense that facial landmarks are used to drive the facial and lip motion, the quantitative results seem to send a mixed signal. In Table 1, LSE-C and LSE-D are only better than MakeItTalk. On CSIM, Wav2Lip is the best instead of the proposed approach. CPBD is also worse than a few state-of-the-arts. On FID, the proposed approach achieved the best score but only marginally better than Wave2Lip. On qualitative results, there are only a handful of examples. It would be more convincing to provide additional examples or even better, to provide videos to show the temporal evolution. Therefore, it is hard to say the results achieved by proposed method is substantially better than state-of-the-arts.
2. The ablation study on proposed approach can also be strengthened to demonstrate the impact of key design choice. For example, on AtoM-Net, the use of local attention on split of facial landmarks was claimed to be an important factor of improving performance. But there is study on how the specific split of landmarks (e.g. number of split, composition of each split) affect the performance. There is no ablation study on MC-VDM. One factor that is worth analyzing is the impact of different loss term used to train the model.
3. It was not clearly explained why proposed method can achieve faster training and inference time.

**Questions:**

1. In MC-VDM architecture illustration Figure 3, does the video encoder takes landmark and image frames jointly as input? Or is it the video encoder will encode landmark and image sequence separately? If encoding was done separately, are we using the same encoder for two types of input?
2. According to the paper, it seems AtoM-Net and MC-VDM are trained separately. Have the authors considered joint training of the two modules? Do we have any expectation or known challenging in doing so?
3. Some notation was not specified correctly or consistently. Can the authors clarify? For example, in the last line of page 5, $l\in \mathbb{R}^{S\times H \times w}$ should be $l\in \mathbb{R}^{S\times H \times W}$. The second line of page 6, should $zr^s$ be $zr^{xy}$? What is the $\lambda$ value used in defining $\mathbb{E}_{\epsilon, t}$ and how is it determined?
4. Finally, there are some typos, for example, in page 8, sentence 'Lip-sync metrics even though we. ' seems to be broken from the context. The authors should proofread the paper.

---

### Official Review · Reviewer_4W8w · 2023-11-01

**Soundness:** 3 good
**Presentation:** 3 good
**Contribution:** 2 fair
**Rating:** 5
**Confidence:** 4

**Summary:**

This paper presents a two stage approach for generating single image talking head videos, based on diffusion networks.  This two stage process consists of: a transformer-based diffusion model that generates facial landmarks from audio (using 3DMMs for frontalization of landmarks), and in the second stage a diffusion net uses the landmarks and a reference image to genrate a naturalistic talking head video.  The authors propose using different attention networks for lip and non-lip face regions, which aims to address the fact that lip variability has less independence from audio, with different dynamics.  The results are promising over prior diffusion models, and this model is more optimized / faster to train than the ones compared with.

**Strengths:**

The idea of separate lip/non-lip attention is valuable and works well, although different variation of head parts could emerge from the design of the architecture rather than a-priori

The results are shown to be state of the art, and the model is efficient

**Weaknesses:**

limitations are not adequately addressed

evaluation metrics are lacking - perhaps not only for this paper but for the problem at hand (E.g. see below)

Ablation studies are focused on AToM-NET but not much on MC-VDM

Comparisons in speed only with Difftalk in table 2?

**Questions:**

the limitations of the method are not adequately mentioned. also, what are the limitations of the metrics employed? most measure lip to audio similarity,

methods that change only lips excel in video quality; but how can one measure that generating the video is in more agreement with the audio rather than simply adapting mouth? (I can see intuitively why this should be more reasonable of course)

what happens when 3dmm fitting fails? how does this affect generalization to more diverse expressions?
r